# Short-Term Effects of Salt Stress on the Amino Acids of *Phragmites australis* Root Exudates in Constructed Wetlands

En Xie [1], Xuejing Wei [2], Aizhong Ding [2,*], Lei Zheng [2,*], Xiaona Wu [2] and Bruce Anderson [3]

1   College of Water Resources and Civil Engineering, China Agricultural University, Beijing 100083, China;
    xe@cau.edu.cn
2   College of Water Sciences, Beijing Normal University, Beijing 100875, China; wjblsfwzxyy@163.com (X.W.);
    wxn19920123@163.com (X.W.)
3   Faculty of Engineering and Applied Science, Queen's University, Kingston, ON K7L 3N6, Canada;
    bruce.anderson@queensu.ca
*   Correspondence: ading@bnu.edu.cn (A.D.); zhengleilei@bnu.edu.cn (L.Z.); Tel.: +86-13910989543 (A.D.);
    +86-13810821932 (L.Z.)

**Abstract:** In this study, the short-term effects of NaCl stress on the free amino acid content and composition of root exudates of *Phragmites australis* were evaluated. Nineteen amino acid types were detected in all samples. The results indicated that NaCl significantly influenced the total amino acid (TotAA) content. The TotAA content at 6‰ salinity (1098.79 μM g$^{-1}$ DW) was up to 24 times higher than that in the control group (45.97 μM g$^{-1}$ DW) but decreased to 106.32 μM g$^{-1}$ DW at 6‰ salinity in the first hour. The stress period also significantly affected the TotAA content. After 4 h of stress, the TotAA content of the control and 1‰ salinity groups increased by approximately 30- and 14-fold, and those of the 3‰ and 6‰ groups decreased to 60% and 37%, respectively. The increase in TotAA content was primarily caused by the increase in proline content; the proportion of proline accounted for 58.05% of the TotAA content at 3‰ salinity level in 2 h. Most amino acids showed a significant positive correlation with each other, but proline and methionine showed a different trend. Therefore, the proline level is a useful indicator of salt stress in *Phragmites australis*, especially in saltwater wetlands.

**Keywords:** *Phragmites australis*; salt stress; root exudates; amino acids; proline

## 1. Introduction

Salt stress is one of the most important environmental stress factors that affect wetland plants [1]. It causes a series of physiological and biochemical changes in plants through osmotic stress, ion toxicity, and imbalance in active oxygen metabolism [2]. For instance, physiological drought, nutrient deficiency, and cell structure damage can occur, and, especially, plant roots can be immediately injured by high salinity [3]. *Phragmites australis*, a representative wetland plant, is found in both freshwater and saline habitats [4] and even adapts to high-salinity regions. Many previous studies have shown that *P. australis* is easily propagated by seed dispersion and vegetatively propagated from vertical and horizontal rhizomes and stolons [5]. These advantages make *P. australis* dominant in many wetland ecosystems. In the worldwide map of *P. australis* by Srivastava [6], the geographical distribution of *P. australis* extends from wetlands in cold temperate regions to tropical wetlands. Furthermore, *P. australis* has a robust ability to mitigate environmental pollution, making it the most preferred plant for application in constructed wetlands [7,8]. *P. australis* could potentially be used for the treatment of saline wastewaters or constructed wetlands in areas with high evapotranspiration, which increases

the salinity of the wastewater [9]. There is a lot of interest in saline wastewater treatment with *P. australis* [10,11].

Previous studies have indicated that the threshold of salt tolerance in *P. australis* varies widely between 5% and 65%, which reflects the ability of *P. australis* to adapt to different environments via its widely varying salt tolerance [12–14]. Mauchamp [15] found that *P. australis* maintained 100% germination rate when affected by salinity of more than 10‰. Fageria [16] reported that salt not only affected the growth and distribution of plant roots but also affected the absorption of nutrients. A large number of studies have shown that *P. australis* can acclimatize to saline conditions by suppressing salt transport to the shoot or salt sequestration or exclusion [17]. Moreover, many plants, including *P. australis*, secrete and accumulate a cocktail of organic compounds (carbohydrates, organic acids, and amino acids) as an adaptive strategy under salinity stress, and the exudation patterns of the plant roots strongly influence plant performance and health [18–21]. Root exudates also play a crucial role in phytoremediation engineering [22].

Amino acids, in addition to being the basic components of all living cells [23,24], are important components of root secretion [25,26]. Amino acids play highly diverse and essential roles in plants. Plants take up amino acids directly from the soil or assimilate inorganic N into amino acids [27]. As the building blocks of proteins and enzymes, amino acids are important components for plant structure and metabolism [28,29]. Amino acids in root secretion have also been regulated to compensate for the effects of plants under stress and play an important role in plant development, growth, and stress resistance [30]. Plants induce the accumulation of compatible substances to adjust the osmotic pressure under salt-stress conditions [31]. Proline, an amino acid, has been found to be an osmotically active organic solute in the root area [32]. Zhang et al. [33] found that the levels of pyrimidine and purine in tobacco decreased when treated with 50 mmol (mM) sodium chloride (NaCl), but the levels of proline and aromatic amino acids increased when the concentration of NaCl was increased to 500 mM. Thus, some previous studies have used amino acid levels as an indicator of salt stress [32,34,35].

Previous studies on the salt tolerance of plants focused on growth and physiological attributes, such as gas exchange, relative leaf water content, and photosynthetic parameters [5,36], or salt transport and ion balance [32,37]. However, little is known about amino acid patterns in relation to salinity; specifically, there is limited information on the effects of salinity stress on the amino acid content in the root exudates of *P. australis.* The examined stress periods had a duration of a few days or even weeks, and short-term stress (e.g., several hours) has been minimally examined. Therefore, the main objective of this study was to analyze the amino acids exuded by the roots of *P. australis* in response to short-term stress due to varying NaCl concentrations. This information aids in understanding the root exudates elicited by *P. australis* in response to salinity stress and acclimation of *P. australis* to different environmental conditions.

## 2. Material and Methods

### 2.1. Collection of Plant Samples

In the *P. australis* wetlands of Yizhuang Town, Daxing District, Beijing, China (Supplementary Materials Figure S1), spring buds of *P. australis* with attached rhizomes were collected in May, and then replanted into three plastic buckets (diameter 37 cm and height 35 cm, to simulate a constructed wetland, the external walls of the buckets were shaded with light-impermeable foil) containing peaty substrates. The initial chemical characteristics of the substrates were as follows: 38% organic matter, 0.9% available N kg$^{-1}$ soil, 0.7% available P (P$_2$O$_5$) kg$^{-1}$ soil, 1% available K (K$_2$O) kg$^{-1}$ soil, pH 6.8, and electrical conductivity (EC) of 2 mS/cm. All cultivars were grown under natural light and uncontrolled temperature conditions. After 6 weeks of potting, plants that appeared unhealthy were discarded, and a bucket of *P. australis* samples of similar size was selected and excavated out of the peaty substrate with care to avoid damage to the roots. The peat was cleaned from the surfaces of the plants with tap water, and the plants were rinsed three times with deionized water.

### 2.2. Salt Stress and Sample Pretreatments

*P. australis* samples from the same bucket were randomly sorted into five treatment groups (five salinity levels) and separately replanted into containers shaded with light-impermeable foil to prevent light penetration from the sides, then, 500 mL of deionized water was added. The salinity level of each container was adjusted using NaCl (AR, Sinopharm, China). NaCl was added to deionize the water to produce salinity levels of 0‰ (blank), 1‰, 3‰, 6‰, and 10‰. These salinity levels were selected based on a previously work [32]. Subsequently, each *P. australis* was maintained in the corresponding containers for 1, 2, and 4 h [38], with three buckets for each treatment group (5 salinity levels × 3 stress periods), each treatment group had two duplicates. Five hundred milliliters of the NaCl solution containing the root exudates was filtered through 0.45 μm membranes (Millipore, Builington, MA, USA) to remove the residual roots and maintained at 4 °C for subsequent analyses.

NaCl in the root exudate samples was removed to prevent damage to the high-performance liquid chromatography (HPLC) column. We used the method described by [39] with some modifications to remove NaCl. The samples were evaporated under reduced pressure at 70 °C to dryness. The residues (primarily NaCl) were dissolved in glacial acetic acid and filtered through a 0.45 μm membrane. The evaporation procedure for the amino acid standards was performed in the same manner as that for the samples, and the recovery rate of each type of amino acid is listed in Table S1. The filtrates were concentrated to dryness in a rotary evaporator (RE-52A; Zhenjie Instruments, China) under reduced pressure at 50 °C, and the final volume of the residual solution was adjusted to 2 mL for derivatization.

### 2.3. Analyses

The samples, including the amino acid standards (18AA; Sigma-Aldrich, St. Louis, MO, USA) and taurine standards (Sigma-Aldrich, St. Louis, MO, USA), were adjusted to the required concentration with deionized water and derivatized with fluorescein isothiocyanate (FITC; Thermo Fisher Scientific, Waltham, MA, USA) to produce fluorescent amino acids. The samples were filtered through 0.45 μm sample filters and run on an HPLC system (Dionex U3000; Dionex Corporation, Acworth, GA, USA) by using a Sentry-C18 8 precolumn (Waters Corporation, Milford, MA, USA) and amino acid column (4.6 × 250 mm; SMA). The sample injection volume was 20 μL, and the flow rate was 1 mL min$^{-1}$. Separation was performed at 40 °C with a variable gradient of 0.1 mol/L sodium acetate (pH 6.5), acetonitrile (93:7, *v/v*), and acetonitrile:water (80:20, *v/v*) (Table S2). Detection (UV detector) was performed at 254 nm. The retention time of each amino acid is listed in Table S3. The total elution time was 55 min. Standard mixtures of the amino acids were used for identification and quantification of the samples. The total free amino acid (TotAA) content was calculated as the sum of all 18 detected and quantified amino acids. The dry weight (DW) was determined by placing the plant root in an oven at 70 °C until it reached a constant weight. The TotAA content of each amino acid type was provided in absolute units (μM g$^{-1}$ DW), and the percentage of each amino acid type was recorded in relative units (% TotAA) and calculated using the following equation:

$$A\% = (A/TotAA) \times 100\%, \tag{1}$$

where A is the content of certain amino acids (μM g$^{-1}$ DW) and TotAA is the total free amino acid content (μM g$^{-1}$ DW).

### 2.4. Statistical Analyses

A calibration curve was created using amino acid solutions with different concentration gradients, and the results are shown in Table S4. The amino acid concentrations in each sample were calculated by peak area. All data are presented as mean ± standard deviation (SD) from three analyses for each duplicate. Analysis of variance of the data was performed using SPSS 19.0 for Windows (SPSS Inc., Chicago, IL, USA) and Excel 2019 (Microsoft Inc., Redmond, WA, USA). Comparisons of the means were performed using Duncan tests with $p < 0.05$ as the criterion for significance. Redundancy analysis

(RDA) was processed and plotted by Canoco 4.5. Correlation analyses and clustering were performed using R (version 3.6.2) and R Studio Desktop.

## 3. Results and Discussion

### 3.1. Variation in Amino Acids under Different Salt Stress Conditions

The recovery of each amino acid was in the range of 84.09% to 116.35% (Table S1); therefore, the evaporation procedure could be used for NaCl removal. Figure 1 shows variations in the concentration of each amino acid under different salinity stress conditions. After 1 h, the TotAA content increased remarkably by up to 24-fold from 0‰ to 6‰ salinity (Figure 2 and Table S5, from 45.97 to 1098.79 $\mu$M g$^{-1}$ DW). The increase in TotAA was primarily due to the accumulation of proline (Figure 3, 484.80 $\mu$M g$^{-1}$ DW for 3‰ and 290.96 $\mu$M g$^{-1}$ DW for 6‰, accounting for 45.40% and 26.48% of the total amino acids, respectively).

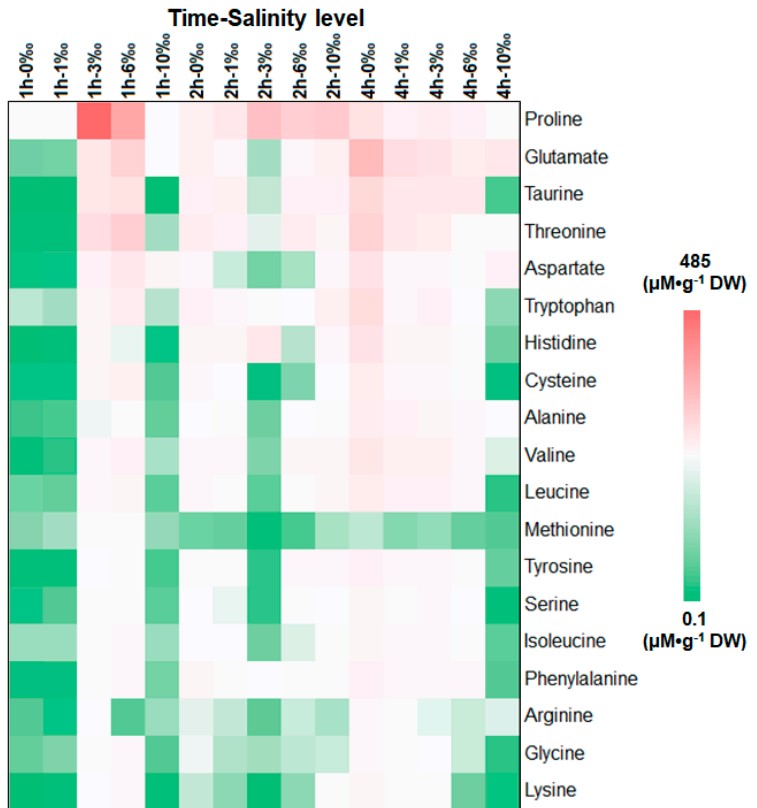

**Figure 1.** Heatmap of variation of each single amino acid with salinity levels and stress time. The red indicates the higher the concentration, and the green indicates the lower the concentration.

The proline content increased up to 36-fold at 3‰ NaCl concentration and 21-fold at 6% NaCl concentration as compared with the control (13.59 $\mu$M g$^{-1}$ DW. Other single amino acids, excluding proline, arginine, and histidine, reached their maximum content at 6‰ NaCl concentration (Figure S2). However, the TotAA and single amino acid content exhibited a sharp reduction from 6‰ to 10‰ salinity levels. Proline exhibited a similar reduction from 3‰ to 10‰ salinity level (Figure 3).

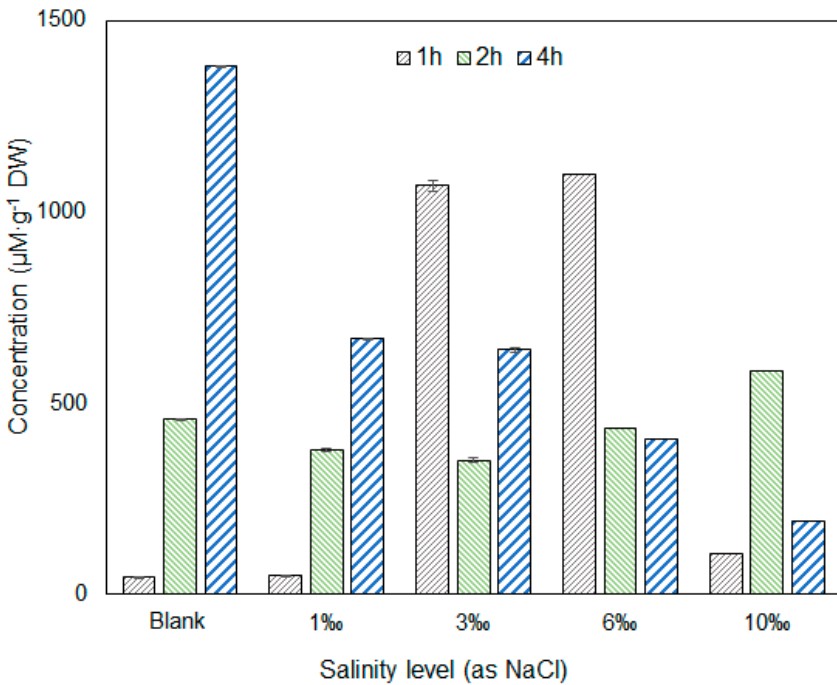

**Figure 2.** Variation of the TotAA with salinity levels. Gray bars represent the concentration of TotAA at one-hour experimental endpoint, green and blue bars represent the concentration at the end of two and four hours. Data presented are the average of two replicates and error bars represent standard deviations of the combined data.

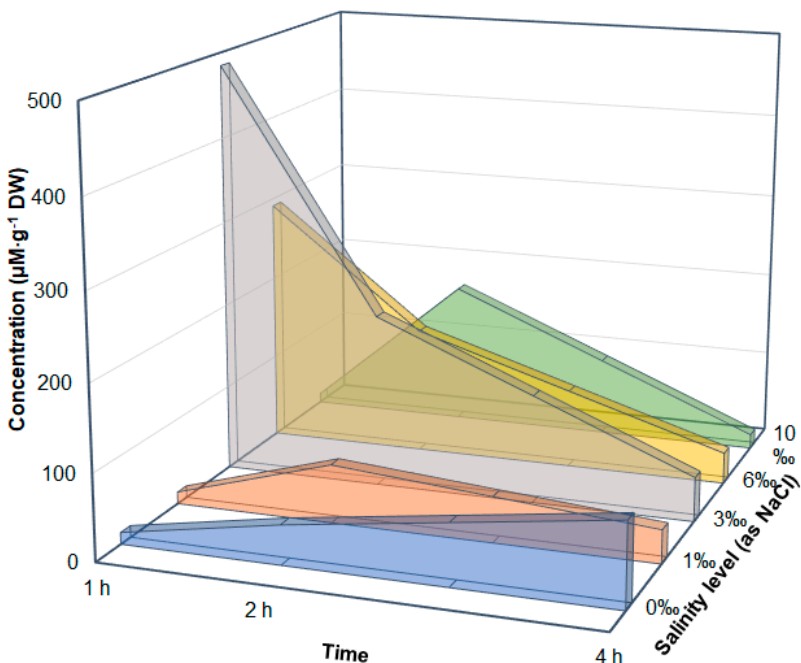

**Figure 3.** Variation of concentration of proline along with time at different salinity stress.

At 2 h of stress, the TotAA content in each treatment group gradually showed similarities (Table S5 and Figure 2). The level of each amino acid type was relatively higher at both 0‰ and 10‰ salinity and the lowest at 3‰ salinity, except for proline and histidine, which showed the highest level at 3‰ salinity and increased by more than four-fold (for proline) and two-fold (for histidine) from 0‰ to 3‰ salinity (Table S5 and Figure S2).

At 4 h of stress, the TotAA content exhibited a tendency to decrease from 0‰ to 10‰ (Table S5 and Figure 2). Each type of amino acid also exhibited a similar trend, i.e., the amino acid concentration decreased as the salinity increased. The levels of all amino acids were the highest at 0‰ salinity, and the levels of almost all of them were the lowest at 10‰ salinity (Table S5 and Figure S2).

Throughout the experimental period, the concentration of most amino acids was higher in the blank sample (Figure S3), except for proline. The proline content was the maximum at 3‰ salinity (Figure 3). Proline dominated the TotAA content, especially in the first 2 h, accounting for 34.40% and 28.86% at 1 h and 2 h, respectively, but only 7.37% at 4 h. At each salinity level, proline accounted for a significant proportion of the TotAA content. In the blank group, proline accounted for only 8.34% of the TotAA content and slightly increased to 11.05% at 0‰ salinity; however, this percentage increased rapidly to 35.97% at 3‰ salinity and, then, was maintained at around 25% at 6‰ and 10‰ salinity (Table S5. The rate of increase in proline content varied substantially at different levels of salinity; the proline content increased by only less than two-fold from 0‰ to 1‰ salinity levels, in contrast to the 36-fold increase from 1‰ to 3‰ salinity levels. The rate of increment in glutamate with time was greater at low salinity levels (0‰ and 1‰) than at high salinity levels (10‰). Thus, we speculated that the proline content increased over a specific NaCl concentration range and decreased when this range was exceeded. This also indicated that proline plays an important role in the adaptation of *P. australis* to salinity stress. In addition, glutamate, threonine, and taurine accounted for a relatively high proportion (8.41% to 14.99%, 6.19% to 11.50%, and 5.27% to 11.26%) of the TotAA content at each salinity level (Table S5).

Furthermore, the TotAA content and most of the single amino acids increased with time at 0‰ and 1‰ salinity levels (Figure 1 and Table S5). The TotAA content exhibited strong responses in the first hour and increased by up to 30-fold in the control and 14-fold at 1‰ salinity level at 4 h. Glutamate increased from 3.1 to 228.03 $\mu M\ g^{-1}$ DW in the blank treatment, i.e., by up to 74-fold within 4 h. At 1‰ salinity level, proline reached its maximum (five-fold higher than the initial content) at 2 h. At 3‰, 6‰, and 10‰ salinity levels, proline content peaked in the first few hours and subsequently decreased substantially.

Overall, during relatively short stress periods, the TotAA content increased. When exposed to a low salinity level, a significant linear relationship was observed between the increase in TotAA content and the stress period. In almost all the treatment groups, proline served as the dominant amino acid; the proportion of proline accounted for up to 58.05% of the TotAA content under 3‰ salinity level at 2 h.

### 3.2. Results of the Correlation and Redundancy Analyses

We also analyzed the correlation between each amino acid. As shown in Figure 4, a significant correlation was observed among all amino acids in general, except proline, methionine, and histidine. In particular, arginine, alanine, phenylalanine, tyrosine, serine, valine, taurine, isoleucine, and leucine showed significant positive correlations ($p < 0.01$). Similarly, a significant positive correlation ($p < 0.01$) was observed among aspartate, glutamate, tryptophan, lysine, threonine, cysteine, and glycine. We clustered all these amino acids into four groups with the hclust package of R (shown as four black wire frames in Figure 4). The first group, proline and methionine, had a significant positive correlation ($p < 0.05$), but no correlation with most of the other amino acids. In the second group, only histidine showed a correlation with about half of the other groups. Then, the rest two groups, as described above, showed a significant positive correlation with each other within the group, respectively.

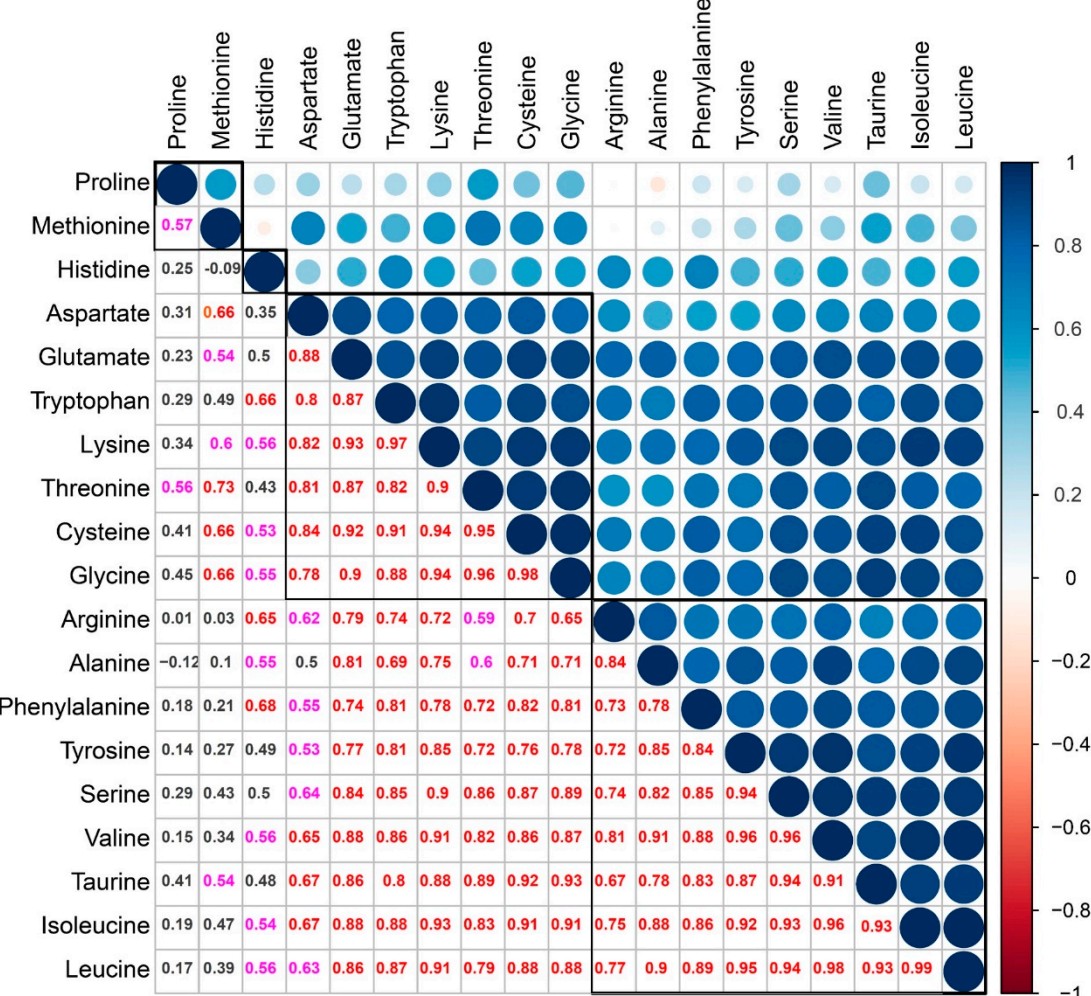

**Figure 4.** Correlation analyses between each amino acid and clustering with hclust. The bubble size and numbers refer to Pearson coefficients, the blue bubble represent positive correlation and red means negative. The purple numbers represent the significant correlation with $p < 0.05$, and the red numbers represent the significant correlation with $p < 0.01$.

Further analysis of the correlations between each amino acid and stress factors (stress period and salinity level) was performed (Figure 5). Redundancy analysis (RDA) was conducted, and the eigenvalues of the first two axes were 0.159 and 0.008. As shown in Figure 5, almost all the amino acids had a positive relationship with the stress period; a negative relationship was found between the stress period and proline or methionine. This indicates that the stress period is more important than the salinity level for the effects of salt stress on the amino acids of *P. australis* root exudates, whether promotion or inhibition. According to the RDA results, the contribution of the stress period was 87.4%, and the contribution of the salinity level was only 12.6%.

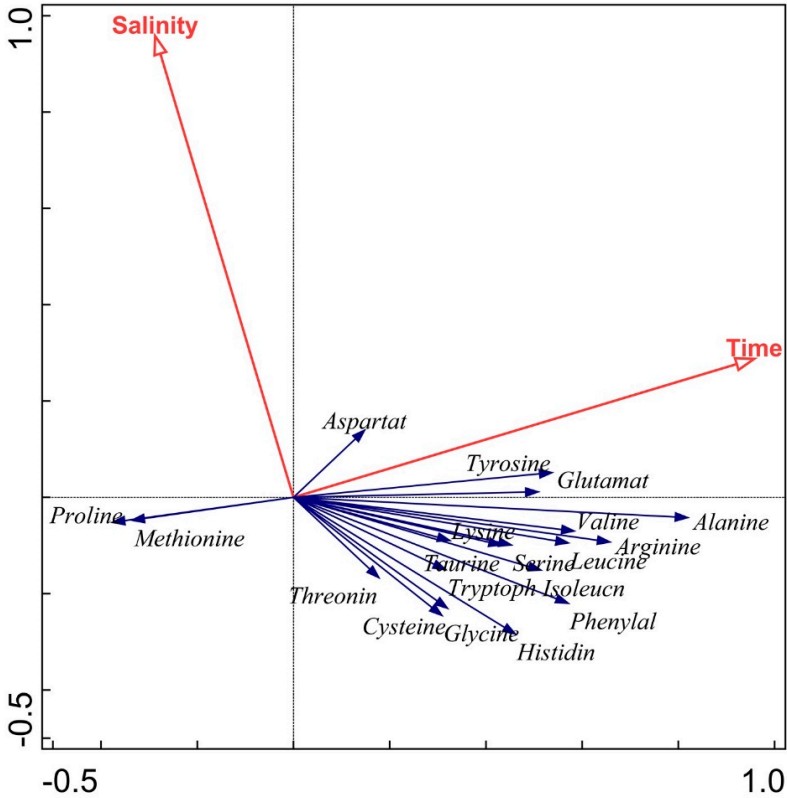

**Figure 5.** Redundancy analysis (RDA) between stress time, salinity level, and amino acids. The red arrows present different stress factors, and the black arrows represent each kind of amino acids.

*3.3. Discussion*

Root exudates are passively released from cells during metabolism, and these exudates are secreted and transported through the cell membrane to the rhizosphere [40]. A large amount of data has shown that plant roots can accumulate amino acids [32,41], particularly proline [42], under high salinity conditions, and these amino acids extravasate to the rhizosphere [43]. Common responses of *P. australis* to salt stressors are changes in the amino acids of the root exudates. Our study indicated that the TotAA content increased within a certain range of stressful conditions. These results are consistent with the findings of Thomas and Hardy [32], who reported the amino acid content in the rhizomes and leaves of *P. australis*.

Comparisons of the single amino acid levels revealed the obvious quantitative importance of proline, which exhibited a dramatic increase with salinity over a specific range. Proline has been extensively studied in the context of plant responses to abiotic stresses. A number of studies have supported the hypothesis that proline accumulation ability and the degree of salinity tolerance are positively correlated in a certain threshold [31,32]. Free proline accumulation seems to be a widespread stress response in higher plants. The pool sizes of several other amino acids also increase under salt stress [44]; however, the degrees of these changes are not comparable to that of proline accumulation, which reaches very high levels within a short period after stress induction [45]. Gzik [46] reported that the amount of proline in sugar beet plants increased under stressful conditions. Changes in the TotAA content are mainly caused by the accumulation of proline in the rhizomes of *P. australis*, as discussed by Thomas and Hardy [32]. Mustapha et al. [5] reported that an increase in the NaCl concentration was associated with an increase in the accumulation of proline. At 200 mM NaCl, the proline concentration was approximately 171% of the control value. According to Yang [36], proline in *P. australis* significantly increased under NaCl salinity stress and exhibited a strong positive correlation with an increase in NaCl concentration.

High concentrations of proline are associated with the protection of membranes and proteins against the adverse effects of inorganic ions. Proline improves the salt tolerance of *Pancratium maritimum* by protecting the protein turnover machinery against stress and damage and upregulating stress protective proteins [47]. Salt stress signals induce the loss of the feedback inhibition of A'-pyrroline-5-carboxylate synthetase (P5CS; a key enzyme in proline biosynthesis [45]), which results in proline accumulation. Proline plays an important role in the acclimation to salinity stress by mediating osmotic adjustments and protecting the subcellular structures of stressed plants [48]. Stress-mediated changes in proline biosynthesis [49], including the hydrolysis of proteins at toxic NaCl concentrations [50], oxidative degradation processes [51,52], reduced incorporation of free amino acids into proteins, and inactivation of proline degradation, can result in increased proline levels in plants exposed to different stresses, including salinity stress [53].

In our experiment, the proline content increased by less than two-fold at 1‰ salinity level as compared with the control; in contrast, a 36-fold increase was observed between 1‰ and 3‰ salinity levels. These findings indicate that a critical salinity level exists. As found in sugar beet, wheat, and other species [54], proline is only slowly accumulated until the plant reaches a critical salinity level. Exceeding this salt concentration results in a greater reaction and accumulation of proline in very high amounts. Therefore, *P. australis* at 1‰ NaCl concentration has not reached its critical salinity level, and 3‰ salt concentration is considered to be the critical salinity level in our study. This explanation is also consistent with the findings of Thomas and Hardy [32].

In our study, the proline level decreased sharply when the salinity level exceeded 3‰. Daniela et al. [30] explained that this tendency to increase the level of free proline inhibits the biosynthesis of excessive amounts of proline in plants. High concentrations of proline can ameliorate the adverse effects of inorganic ions, and this mechanism is associated with the protection of membranes and proteins. The rapid catabolism of proline after relief from stress provides reducing equivalents that support mitochondrial oxidative phosphorylation and generation of ATP for the recovery from stress and repair of stress-induced damage. Therefore, proline levels and proline accumulation dynamics can be used as indicators of salt stress in *P. australis*.

Moreover, our study revealed a positive correlation between the amino acid content of the control group and the stress period. All 19 types of single amino acids were accumulated as the stress period increased, and these amino acids reached their peak concentrations with the longest stress period. This phenomenon is consistent with the *Lycopersicon esculentum* results [55] for the control group, and amino acids and proline in wild species increase as the duration of stress period increases.

This study was performed on a laboratory scale; thus, some details will inevitably be missing as compared with those in a natural wetland. In particular, the presence of wetland substrate can change the way the plants respond to environmental stress and can impact the characteristics of rhizospheric secretion. In addition, salinity stress is not caused by only NaCl; therefore, the more complex interactions between salt levels and amino acid secretion in the natural environment should be noticed. Although there is a certain degree of correlation between amino acid content in both plant tissues and root secretion, further investigation, such as the direct measurement of amino acid content in plant tissues, is essential to quantify the relationship.

## 4. Conclusions

Root secretion plays an important role in plant growth, rhizosphere ecological regulation, and biogeochemical cycle of nutrient elements. The variations and trends in the rhizosphere microenvironment can be judged by the composition and content of root exudates. Amino acids are an essential component of root secretion. Therefore, it is important to discuss the trend of each component of root exudates, especially amino acids, under different environmental stress conditions.

In this study, the results indicated that salinity stress caused by NaCl significantly influenced *P. australis* and the effects of salinity on specific free amino acids played an important role in the adaptation of the plant to salt stress. The TotAA content of the *P. australis* root exudates exhibited significant variations

when exposed to different levels of salt, as well as the stress periods. Therefore, it is important to investigate the effects of salt stress on each type of amino acid. The results showed that the amino acids generally had a significant positive correlation, but they could still be clustered into four groups.

Among them, proline was the dominant amino acid and showed relatively unique trends. Proline is a useful indicator of salt stress, which can monitor the health state of the plant directly, avoiding inaccurate information resulting from some indirect function when only salinity is used to characterize the salt stress in complex sewage treatment. Thus, it is important to discuss the trend under different salinity stress levels when adopting the *Phragmites australis* wetland system for salinity wastewater treatment or phytoremediation of salinized soil. It is important to understand the stable operation and functional guarantee of the *Phragmites australis* wetland system.

**Supplementary Materials:** The following are available online at http://www.mdpi.com/2073-4441/12/2/569/s1, Figure S1: Spring buds of *P. australis* Sampling site in wetland of Daxing District, Beijing, Figure S2: Changes of amino acids in different treatments in each period. Defined the highest in each row as 100%, the rest of each row is converted to percentage by relative value, Figure S3: Changes of amino acids in different treatments throughout all the experimental period. Defined the highest in each row as 100%, the rest of each row is converted to percentage by relative value, Table S1: The recovery rate of NaCl remove procedure, Table S2: The gradient elution process of amino acid mobile phase A and B, Table S3: The HPLC elution time for amino acids, Table S4: The calibration curve for each kind of amino acid and relative standard deviation (RSD), Table S5: Content of total free amino acids (TotAA) and single compounds (absolute content) of root exudates of *Phragmites australis* clones (stress time 1, 2, 4h) compares for five NaCl-salinity levels.

**Author Contributions:** L.Z. and A.D. designed the study; E.X. and X.W. (Xuejing Wei) conducted the experiments and interpreted the data; E.X. and X.W. (Xiaona Wu) contributed to data analysis; E.X. and X.W. (Xuejing Wei) wrote the manuscript; A.D., L.Z., and B.A. revised the manuscript. All authors have read and agreed to the published version of the manuscript.

**Funding:** This research was funded by the National Natural Science Foundation of China (grant number 41203060 and 51509003) and the Fundamental Research Funds for the Central Universities (grant number 2013YB61).

**Acknowledgments:** We are grateful and appreciate Prof. Bruce Anderson for the valuable language polishing of this manuscript. We would like to thank Editage (www.editage.cn) for English language editing.

**Conflicts of Interest:** The authors declare no conflict of interest in relation to the work.

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
