# Peer review of "Short-Term Effects of Salt Stress on the Amino Acids of Phragmites australis Root Exudates in Constructed Wetlands"

_water, doi:10.3390/w12020569_

Round 1
Reviewer 2 Report
Authors present an interesting work on a relevant topic. In fact, salt stress is very relevant either in phytoremediation or in the use of constructed wetlands. Moreover, Phragmites australis is one of the most commonly used plant. So it is very relevant to understand its response to salt stress in order to fully address its utility to remediate salty water/wastewater.
Some revisions are need before the manuscript can be accepted. Please, see below.
In general, English language needs revision.
Abstract
It should also indicate that time had a significant effect on salinity effect. And it should be a take home message at the end of the abstract. Why are these results relevant? Why they should be taken in consideration when using Phragmites australis, namely in constructed wetlands or phytoremediation?
Introduction
Line 44-45 – rephrase the sentence, it is not clear what authors mean
Line 62-64 - rephrase the sentence, English not correct, sentence not clear!
Line 66-67 - revise, not clear! proline is an aminoacid? proline is a osmotically active organic solute? where proline can be found?
Line 69 - was up to 500 nM? Please, revise the English
Line 70 - adopted or showed?
Material and Method
Section 2.1 – line 92 - what do you mean by natural environment? Please, add relevant information
Section 2.2 –
how many plants per container?; why were these levels of salinity selected?; only one container per condition?; Please, clarify and add relevant information
lines 108-114 – any test with standard solution to evaluate the recovery of the aminoacids subjected to this evaporation procedure?
line 113 - 2 ml of what solution. Please, clarify and add relevant information
Section2.3 – this HPLC method was optimised by the authors? If so, what was the precision of the methodology? Aminoacid standards were prepared in deionised water? External calibration for quantification or standard addition? Did you doped any sample to confirm the recovery of the aminoacid and evaluate the suitability of the calibration used for aminoacid quantification? Authors mention, aminoacid identification, but since a HPLC was used, and not a LC-MS, I suppose the identification was done by matching retention time of each aminoacids. If this is case it should be indicated in the text.
Line 132-133 – authors say “The contents are given in absolute (μM g -1 DW) and 132
relative (% TotAA) units” – content of what? How was relative TotAA calculated? Please, clarify and add relevant information
Section 2.4 – line 137 – authors indicate that the mean and standard deviation was calculated, but it is not clear form previous section how many replicates were there for treatment. So, this is the mean of which samples? Please, clarify and add relevant information
Results
Line 145 – “different not “deferent”
Figure 2 caption – please, indicate that the values being shown are the average of 3 replicates with the respective standard deviation. The same for other figures caption. Why some values do not present the standard deviation? In xx axis instead of treatment indicate Salinity (as NaCl). The same for all figures.
Fig 3 - “different not “deferent”
Discussion
Line 239-245- add relevant references to these sentences
Line 249- this reference should be a number
Line 253-254- references needed
Line 260-265 – references should be in numbers
Line 289 – reference number is missing
Line 301- reference is missing
Discussion should also mention that these experiments were carried out in a very simplified way in which the plants roots were in direct contact with the saline water. In case of constructed wetlands, the presence of a substrate can have an effect on the response of the plants to salinity. Moreover, the medium used for the tests only had NaCl and salinity normally results from the combination of several salts. So, authors must consider the simplicity of the medium in the response observed.
Authors’ discussion a lot accumulation of aminoacids in plant tissues, but in this experiment, authors only measured aminoacids released by the plants. They did not measured aminoacids directly in plant tissues. So, the correlation between aminoacids in solution and the aminoacids accumulated in plant tissues is not straightforward and that should also be included in Results discussion
Discussion or conclusion should have a take home message, as I mention to the abstract. Why are these results relevant? Why they should be taken in consideration when using Phragmites australis, namely in constructed wetlands or phytoremediation?
